# Mesoporous Organosilica Nanoparticles with Tetrasulphide Bond to Enhance Plasmid DNA Delivery

**DOI:** 10.3390/pharmaceutics15031013

**Published:** 2023-03-22

**Authors:** Yue Zhang, He Xian, Ekaterina Strounina, Kimberley S. Gunther, Matthew J. Sweet, Chen Chen, Chengzhong Yu, Yue Wang

**Affiliations:** 1Australian Institute for Bioengineering and Nanotechnology, The University of Queensland, Brisbane, QLD 4072, Australia; 2Centre for Advanced Imaging, The University of Queensland, Brisbane, QLD 4072, Australia; 3Institute for Molecular Bioscience (IMB), IMB Centre for Inflammation and Disease Research, and Australian Infectious Diseases Research Centre, The University of Queensland, Brisbane, QLD 4072, Australia; 4School of Biomedical Sciences, Faculty of Medicine, The University of Queensland, Brisbane, QLD 4072, Australia

**Keywords:** mesoporous materials, organosilica, tetrasulphide bond, DNA transfection, dendritic cell

## Abstract

Cellular delivery of plasmid DNA (pDNA) specifically into dendritic cells (DCs) has provoked wide attention in various applications. However, delivery tools that achieve effective pDNA transfection in DCs are rare. Herein, we report that tetrasulphide bridged mesoporous organosilica nanoparticles (MONs) have enhanced pDNA transfection performance in DC cell lines compared to conventional mesoporous silica nanoparticles (MSNs). The mechanism of enhanced pDNA delivery efficacy is attributed to the glutathione (GSH) depletion capability of MONs. Reduction of initially high GSH levels in DCs further increases the mammalian target of rapamycin complex 1 (mTORc1) pathway activation, enhancing translation and protein expression. The mechanism was further validated by showing that the increased transfection efficiency was apparent in high GSH cell lines but not in low GSH ones. Our findings may provide a new design principle of nano delivery systems where the pDNA delivery to DCs is important.

## 1. Introduction

Introducing deoxyribonucleic acid (DNA) molecules into cells has garnered extensive attention in various applications, such as gene editing [1,2], infectious disease treatment [3,4], and cancer immunotherapy [5]. For example, in cancer immunotherapy, dendritic cells (DCs) are the most potent antigen presenting cells and are thus attractive target cells for DNA transfection [6]. Due to the impermeability of the cell membrane against large negatively charged DNA [7], various systems have been applied for DNA delivery into cells [8,9]. Compared to viral vectors with safety concerns, nanomaterial-based non-viral vectors have been widely studied as a promising gene delivery system due to their ease of preparation and better biocompatibility [10,11]. Despite the immense progress in the research of nanomaterial-based gene delivery systems, efficient delivery of DNA into hard-to-transfect cells (for example, dendritic cells) remains challenging due to low transfection efficiency and protein expression level [12]. Current DNA delivery systems mainly focus on delivery function modulation, such as tuning nanoparticle size to enhance cellular uptake [13,14] and adjusting nanotoprography to prevent enzymatic degradation [15]. However, few reports have investigated the impact of nanoparticles on translation regulation.

Mesoporous organosilica nanoparticles (MONs) have been widely studied due to their tuneable mesostructures and variable compositions [16,17,18]. In particular, tetrasulphide bonds of MONs have been applied to deplete glutathione (GSH) [19], thus promoting the mammalian target of rapamycin complex 1 (mTORc1) pathway activation and enhancing mRNA translation in high-GSH DCs [20,21]. Wu et al. used tetrasulphide-bridged MONs for plasmid DNA (pDNA) transfection in Hela (cancer) cells [22] and revealed that large radial pores (6.2 nm) enabled MONs with higher pDNA loading capacity and better protection from enzymatic degradation, leading to enhanced pDNA transfection efficacy [22]. However, previous reports limited the research on the effect of MON structural difference on pDNA transfection. The effect of tetrasulphide bonds within MONs on pDNA transfection and translation efficacy is to be investigated.

Herein, we firstly studied the impact of tetrasulphide bonds within MONs on pDNA transfection and translation in DCs with high GSH levels. The tetrasulphide content consumed the intracellular GSH to activate the mTORc1 pathway, leading to ribosomal protein S6 (RPS6) phosphorylation. This enhanced translation and pDNA transfection efficacy. Compared to inorganic mesoporous silica nanoparticles (MSNs) and commercial transfection reagents (lipofectamine), tetrasulphide-bridged MONs enhanced pDNA transfection in DCs. This study provides a new design principle for nanomaterial-mediated pDNA transfection in DC.

## 2. Materials and Methods

### 2.1. Materials and Reagents

Triethanolamine (TEA, >99.5%), cetyltrimethylammonium bromide (CTAB), tetraethyl orthosilicate (TEOS, 98%), tetrabutyl orthosilicate (TBOS, 97%), bis(triethoxysilyl)propane tetrasulfide (BTES), sodium 3-(trihydroxysilyl) propylmethylphosphonate (THPMP), polyethyleneimine (PEI, branched, molecular weight: 10,000), phosphate buffer (PBS, 10 mM, pH 7.4)), fetal bovine serum (FBS), dimethyl sulfoxide (DMSO), sodium hydroxide (NaOH, ≥98%), 4′,6-diamidino-2-phenylindole (DAPI), and bovine serum albumin (BSA) were purchased from Sigma-Aldrich (Castle Hill, NSW, Australia). Sodium heptafluorobutyrate (FC4) was purchased from Santa Cruz Biotechnology (Santa Cruz, CA, USA). Dulbecco’s modified eagle medium (DMEM), Gibco Roswell Park Memorial Institute (RPMI) 1640 medium (ATCC modification), and trypsin-EDTA (0.25%) were purchased from Life Technologies (Australia). The following materials, 5,5′-dithiobis(2-nitrobenzoic acid) (DTNB), 3-(4,5-dimethylthiazol-2-yl)-2,5-diphenyltetrazolium bromide (MTT), and lipofectamine 2000, were purchased from Thermofisher Scientific (Australia). The cellular GSH detection assay kit and Alexa fluor 647 conjugated phospho-S6 ribosomal protein (p-RPS6) antibody were purchased from Cell Signaling Technology (Notting Hill, VIC, Australia). The OVA enzyme-linked immunosorbent assay (ELISA) kit was purchased from Biomatik (Wilmington, DE, USA). pDNA encoding enhanced green fluorescent protein (EGFP) and ovalbumin (OVA) were purchased from the BASE Facility (The University of Queensland, QLD, Australia). DC 2.4 cells and HEK293T cells were purchased from American Type Culture Collection (Manassas, VA, USA).

### 2.2. Synthesis of MONs/MSNs

In a typical synthesis, 34.0 mg of TEA was dissolved in 12.5 mL of deionized water for stirring at room temperature for 0.5 h. An amount of 190.0 mg of CTAB and 24.6 mg of FC4 were then added to the above solution and stirred at 80 °C. After 1 h, the premixed inorganic precursor of 0.60 mL of TEOS and 0.80 mL of TBOS was added for stirring at 80 °C for 0.5 h, followed by the addition of 0.60 mL of BTES for further stirring for 24 h. Products were collected by centrifugation through washing with ethanol for three times and dried in the vacuum oven at room temperature overnight. After removal of surfactants through acid extraction (using HCl and ethanol mixed solution at a volume ratio of 1:20) for 6 h at 60 °C for three times or through calcination at 550 °C for 5 h, the final products were obtained, denoted as MONs or MSNs, respectively.

### 2.3. PEI Modification of Nanoparticles

An amount of 30 mg of nanoparticles (NPs, either MONs or MSNs) were dispersed through sonication in 10 mL of pH = 10 water (adjusted by 25% ammonia solution, Sigma-Aldrich), then mixed with 10 mL of 56 mM THPMP solution with stirring at 40 °C. After 2 h, samples were collected through centrifugation and washed with deionized water twice. The product was then resuspended in 15 mL of carbonate buffer containing 30 mg of PEI (10 K) with further stirring for 2 h at room temperature. The final products were collected through centrifugation, washing with deionized water, and subjected to freeze drying. The final products were denoted as PEI-NPs.

### 2.4. Characterisation

Transmission electron microscopy (TEM) images were taken using a JEOL 1010 operated at 100 kV. For TEM characterization, samples were dispersed in ethanol via ultrasonication, then dried on the carbon film on a copper grid. Nitrogen sorption analysis was conducted by a Micromeritcs Tristar II 3020 system at 77 K. The samples were degassed at 393 K for 12 h and 453 K for 6 h for MONs and MSNs, respectively, under a vacuum before measurement. The Barrett–Joyner–Halenda (BJH) method was used to calculate the pore size of samples from the adsorption branches of the isotherms, and the Brunauer Emmett–Teller method was utilized to calculate the specific surface areas. The total pore volume was derived from the adsorbed volume at the maximum relative pressure (P/P_0_) of 0.99, which attenuated total reflectance Fourier transform infrared spectroscopy (FTIR) analysis was conducted on a Thermo Nicolet Nexus 6700 FTIR spectrometer equipped with Diamond ATR Crystal. A solid-state Bruker Advance III spectrometer was used for ^13^C cross-polarization magic-angle spinning (CPMAS) and ^29^Si magic angle spinning (MAS) nuclear magnetic resonance (NMR) spectra with 7T (300 MHz for ^1^H) magnet, Zirconia rotor, 4 mm, rotated at 7 kHz. The zeta potential was measured three times using a Zetasizer Nano instrument by dispersing particles into deionized water under sonication.

### 2.5. Degradation Test at Stimulated Conditions

An amount of 1.5 mg of nanoparticles (MONs or MSNs) were dispersed in 1 mL of PBS buffer containing 10% FBS and various concentrations of GSH (1 mM and 10 mM) and stirred at 37 °C for 48 h. Then, aliquots were taken and washed with PBS before TEM imaging.

### 2.6. GSH Oxidation Assay

The GSH oxidation capability of MONs was evaluated by the reported Ellman test [23], by quantifying the generated thiol groups available on MONs. Specifically, 1.5 mg of MONs were dispersed in 1 mL of PBS with 10% FBS and 1 mM or 10 mM GSH. After stirring at room temperature for 48 h, the suspension was centrifuged, and the precipitate was collected. After washing with potassium phosphate buffer (pH = 8) for three times, the pellet was resuspended in 1 mL of potassium phosphate buffer (pH = 8). An amount of 1 mL of 10 mM DTNB solution (in potassium phosphate buffer, pH = 8) was then added to the 1 mL of suspension and stirred at room temperature for 10 min. The suspension was then centrifuged, and the coloured supernatant was collected for the UV/vis measurement at 412 nm.

### 2.7. pDNA Loading

For loading of pDNA-encoding EGFP, 2.5 μL of PEI-NPs (2 μg/μL) was mixed with 1 μL of pDNA (1 μg/μL) in 10 μL of 10 mM PBS solution for incubation at 4 °C for 30 min. The mixture was then centrifugated at 15,000 rpm for 5 min. The supernatant was collected for determining the amount of pDNA using a Nanodrop 1000 spectrophotometer (Thermo Scientific) at a wavelength of 260 nm (PBS as the blank). The loaded amount of pDNA on PEI-NPs was calculated by the difference between the original and residue amount of pDNA in the supernatant. All experiments were performed in duplicate.

### 2.8. Cell Viability Assay

MTT assays were used as an indirect read-out of DC2.4 cell viability. Cells were seeded in a 96-well plate at a density of 9000 cells per well. After 24 h of incubation in the 37 °C, 5% CO_2_ incubator, the culture medium was replaced with 100 µL of fresh medium containing various concentrations of PEI-NPs (10, 20, 40 μg/mL) and incubated for 24 h. Then, 10 μL of MTT solution (5 mg/mL) was added to each well for another 4 h incubation. The medium was removed before 200 μL of dimethyl sulfoxide (DMSO) was added to each well to measure the absorbance at 540 nm using a microplate reader. The cell viability was determined using cells treated with PBS as the control. All experiments were performed in triplicate.

### 2.9. Cellular Uptake Assay

The cellular uptake of PEI-NPs was studied in DC2.4 cells using an inductively coupled plasma-optical emission spectrometer (ICP-OES). Cells were seeded in a 6-well plate at a density of 1.3 × 10^5^ cells/mL. After 24 h of incubation in a 37 °C, 5% CO_2_ incubator, PEI-NPs were added to each group at a final concentration of 50 μg/mL for 4 h. Cells were then washed with warm PBS and harvested by centrifugation. The cell numbers for each well were counted. Afterwards, 120 μL of deionized water was added to each sample with sonication for 2 h to lyse the cells. The samples were then centrifugated at 15,000 rpm for 10 min. The precipitates were dried in a 100 °C oven overnight. Samples were finally dissolved in 150 μL of 1 M NaOH solution and shaken at room temperature for 24 h before ICP-OES quantification. All experiments were performed in triplicate.

### 2.10. In Vitro pDNA Transfection Assay

The in vitro transfection of EGFP-pDNA by PEI-NPs was evaluated in DC2.4 cells and HEK293T cells. The commercial transfection agent, Lipofectamine 2000 was used as a positive control, with the recommended dosage. Cells were seeded into 12-well plates at a density of 1.3 × 10^5^ cells per well for 24 h incubation before transfection. An amount of 1 μg of pDNA was mixed with 40 μg of PEI-NPs in 50 μL of 10 mM PBS solution for 30 min at room temperature, then added by droplets into each well for 48 h incubation. Cells were then harvested to determine the intracellular EGFP expression through flow cytometry. For confocal microscopy visualisation after in vitro transfection, cells were seeded with sterilized glass coverslips in each well with the same density as used for flow cytometry analysis. After 48 h incubation with pDNA and NP mixtures, coverslips were collected and stained with DAPI for confocal imaging (Leica SP8). Ovalbumin (OVA)-expressing pDNA (OVA-pDNA) was also used to determine the intracellular OVA expression after incubation with PEI-NPs. Following the same protocol of EGFP pDNA transfection as above, cells were harvested after 48 h incubation and lysed. OVA levels in lysates were evaluated using an OVA ELISA kit, according to the manufacturer’s protocol. All experiments were performed in duplicate.

### 2.11. GSH and p-RPS6 Measurement

The GSH and p-RPS6 level measurement was conducted in DC2.4 cells after being treated with PEI-NPs. For GSH measurement, cells were seeded in the black 96 well with a density of 1.2 × 10^4^ cells per well. After 24 h of incubation, the culture medium was replaced with a fresh one containing PEI-NPs at a concentration of 40 μg/mL. After 12 h of incubation, 10 μL of live cell staining working solution was added to each well for another 30 min incubation. The fluorescence was measured at an excitation wavelength of 380 nm and an emission wavelength of 460 nm, based on the manufacturer’s protocol. The relative GSH amount was calculated using cells treated with PBS as a control. For p-RPS6 evaluation by flow cytometry, cells were seeded in 12 well plates at a density of 1.3 × 10^5^ cells/mL for 24 h incubation. Then, 20 μL of PEI-NPs (2 μg/uL) was added to each well with further incubation of 24 h. Cells were then harvested, fixed with 4% formaldehyde, and permeabilizated in 90% methanol. Afterwards, cells were stained in diluted Alexa-fluor 647 conjugated p-RPS6 antibody solution (1:50, 18 diluted in 0.5% BSA solution) and washed with PBS for flow cytometry. For immunofluorescence, cells were seeded with coverslips within 12 well plates with the same density for 24 h. The same amount of PEI-NPs was added to each well for further incubation of 24 h. Coverslips were processed in 4% formaldehyde and blocking buffer, then stained in diluted antibody solution (1:100, diluted in 0.5% BSA solution) overnight, washed with PBS, and stained in DAPI for confocal imaging. All experiments were performed in duplicate.

### 2.12. Statistical Analysis

Data are presented as mean ± standard error of the mean (SEM) with at least *n* = 3. The significance between the two groups was analyzed via an unpaired *t*-test, while a multiple-group comparison was analyzed via one-way ANOVA multiple comparisons, using GraphPad 19 Prism 9 software. A *p*-value > 0.05 is considered statistically not significant (ns: *p* ≥ 0.05). Statistically significant levels are defined as *: *p* < 0.05; **: *p* < 0.01; ***: *p* < 0.001; ****: *p* < 0.0001.

## 3. Results and Discussion

MONs and MSNs were synthesised following a reported protocol, using CTAB and FC4 as surfactant templates in the alkaline aqueous system at 80 °C, BTES as the organosilica precursor, while TEOS and TBOS as the inorganic silica precursors [24]. In the reaction, both inorganic and organosilica precursors were hydrolyzed to provide Si-OH groups, which further assembled with the cationic head groups of the surfactant templates [25]. Surfactants were removed through acid extraction and calcination for MONs and MSNs, respectively. TEM images (Figure 1A,B) demonstrate that both MONs and MSNs exhibited similar dendritic mesoporous structures as reported in the literature [18]. By measuring 100 particles from TEM images, the average particle diameters of MONs and MSNs were determined to be 61.5 ± 2.7 nm and 62.2 ± 4.2 nm, respectively. Surfactant removal from MONs was characterized by Fourier transform infrared spectroscopy (FTIR) analysis before and after extraction (Appendix A). Compared to the as-synthesized particles, two characteristic peaks at 2850 and 2918 cm^−1^ originating from methylene groups of CTAB in the as-synthesized MONs were not observed in the acid-extracted MONs, demonstrating successful surfactant removal [26].

To characterize the pore structures of MONs and MSNs, the nitrogen sorption analysis was conducted. Both nanoparticles showed the typical type IV isotherms (Figure 1C). Two major capillary condensation steps were observed for both MONs and MSNs, one at a relatively high relative pressure (P/P_0_) between 0.6–0.9, another at >0.95. The first step indicated the existence of large mesopores [27], while the other should be attributed to interparticle packing voids. The corresponding pore size distribution curves of two nanoparticles were shown in Figure 1D. Both nanoparticles showed peaks centred at 6.9 nm, indicating similar mesopore sizes for MONs and MSNs.

The composition of MONs was characterized by solid-state ^13^C CPMAS NMR and ^29^Si MAS NMR analysis. The results were shown in Figure 2A,B. In the ^13^C NMR spectrum (Figure 2A), three characteristic peaks were observed, corresponding to ^1^C, ^2^C, and ^3^C carbon species in -Si-^1^CH_2_-^2^CH_2_-^3^CH_2_-S-S-S-S-^3^CH_2_-^2^CH_2_-^1^CH_2_-Si-, indicating the successful incorporation of organosilica inside the framework of MONs [28]. Peaks associated with carbon atoms from CTAB molecules were not evident in the ^13^C NMR spectrum, also indicating the successful removal of CTAB [23]. In addition, the presence of T^2^ [C-Si(OSi)_2_(OH)] and T^3^ [C-Si(OSi)_3_] generated from BTES in the ^29^Si NMR spectrum further confirms the existence of bridged organosilica within MONs [28], while T^2^ and T^3^ characteristic peaks were not observed in the ^29^Si NMR spectrum of MSNs (Appendix A), demonstrating the successful removal of organosilicon content during the high temperature calcination process.

Next, the GSH responsive degradability of MONs and MSNs was studied at 1 mM and 10 mM GSH solutions in the presence of serum, equivalent to the intracellular conditions of normal cells and dendritic cells, respectively. As shown in Appendix A, after incubation of MONs with 1 mM GSH solution, partially collapsed structures were observed while the spherical structure was mostly maintained. However, when incubated with 10 mM GSH solution, the structure of MONs was seriously destructed and even partially disintegrated into small debris after 48 h (Appendix A). In comparison, MSNs remained the initial structures at 48 h in both GSH concentrations (Appendix A). The results above showed the GSH-responsive degradability of MONs. Moreover, the capability of MONs to oxidize GSH was evaluated by GSH oxidation assay. After incubation of MONs in serum with 1 mM and 10 mM GSH concentrations, the relative amount of thiol groups as the reduction product on MONs was measured using the Ellman test by determining the optical density (OD) value at 412 nm. As shown in Appendix A, the OD value of MONs incubated with 10 mM GSH solution (0.80) was significantly higher than that of MONs in 1 mM GSH solution (0.67), in accordance with a higher content of thiol groups generated from tetrasulphide bonds within MONs at an increased GSH level.

To enhance pDNA loading and delivery, PEI with a highly positive charge and effective proton sponge effect was modified on nanoparticles for enhancing their pDNA loading and endosomal escape capacity [29]. Both MONs and MSNs were modified with the branched PEI with a molecular weight of 10 kDa. The obtained samples were named as PEI-MONs and PEI-MSNs. As shown from TEM images (Appendix A), PEI-modified MONs and MSNs retain similar particle sizes and pore structures to unmodified ones. To characterize the PEI amount, an elemental analysis was conducted, and the results were shown in Appendix A. The nitrogen weight percentages of PEI-MONs and PEI-MSNs were similar (5.8% and 5.7%, respectively), suggesting similar contents of PEI modified on two nanoparticles.

To study the pDNA loading capacity of nanoparticles, EGFP pDNA was used. Negative charged pDNA was absorbed with positively charged PEI via electrostatic attraction [30]. PEI-MONs exhibited an EGFP pDNA loading capacity of 56.5 ng·μg^−1^ (Figure 3A), slightly higher than that of PEI-MSNs (50.2 ng μg^−1^). The surface charges of nanoparticles were measured before and after pDNA loading (Figure 3B). Bare MONs and MSNs exhibited negative charges of −15.1 and −22.5 mV, while surface charges of both PEI-modified nanoparticles shifted to positive of 16.8 and 15.6 mV, suggesting that the PEI modification was successful. After pDNA loading, PEI-MONs and PEI-MSNs were both negatively charged (−9.6 and −15.3 mV respectively), thus verifying successful pDNA loading. The higher surface charge of MON groups than MSN groups was presumably due to the increased organosilica content and thus decreased inorganic silica on the surface [24], also leading to slightly increased EGFP pDNA loading capacity.

To explore the effect of tetrasulphide on pDNA transfection, DC2.4 cell line was selected. Firstly, to evaluate potential cytotoxicity of nanoparticles, MTT assays were used as a read-out of DC2.4 cell viability after incubation of PEI-NPs at a concentration from 10 µg mL^−1^ to 40 µg mL^−1^ for 24 h. MTT assay data (Appendix A) demonstrated a dose-dependent trend towards reduced cell viability, with >65% cells remaining viable even at the highest concentration of 40 µg mL^−1^ for both PEI-MONs and PEI-MSNs. This concentration was selected for later cell culture experiments. Then, the cellular uptake of PEI-NPs was evaluated after incubation with cells after 4 h through ICP-OES. As shown in Appendix A, cells incubated with PEI-MONs and PEI-MSNs exhibited a similar Si content of 45.3 pg Si/cell and 45.7 pg Si/cell, indicating the similar cellular uptake of the two nanoparticles.

Next, to evaluate the gene delivery efficacy of PEI-MONs/MSNs, EGFP pDNA was used in the in vitro transfection assay in DC2.4 cells. Lipofectamine 2000, a commercial transfection reagent, was used as a positive control. Confocal microscopy was used to visualise EGFP expression in cells after incubation with pDNA/PEI-NPs complexes for 48 h. The cell nucleus was stained with DAPI (blue fluorescence). For PBS and bare pDNA groups (Appendix A), neglectable green fluorescence (representing expressed EGFP) was observed. In contrast, pDNA/PEI-MONs-treated cells showed the highest and strongest green fluorescence (representing expressed EGFP) around the cell nucleus (Figure 4A) compared to pDNA/PEI-MSNs (Figure 4B), pDNA/lipofectamine (Figure 4C). The EGFP expression level in DC2.4 cells was also analysed by flow cytometry. As shown in Figure 5D, pDNA/PEI-MONs-treated cells had the highest transfection efficiency with a positive cell percentage of 49.8% compared to pDNA/PEI-MSNs (38.7%) and pDNA/lipofectamine (9.4%). Moreover, cells treated with pDNA/PEI-MONs also showed the highest mean fluorescence intensity (MFI), which was 2.1 and 2.6 times higher than that of cells treated with pDNA/PEI-MSNs and pDNA/lipofectamine, respectively. 

It was noted that compared to PEI-MSNs, PEI-MONs with tetrasulphide bonds exhibited significantly improved EGFP transfection capacity as evidenced by both confocal and flow cytometry results. To understand the mechanism of the higher pDNA delivery efficacy promoted by MONs, the intracellular GSH and p-RPS6 levels of DC2.4 cells after incubation with PEI-NPs were evaluated and compared to the PBS control group. As shown in Figure 5A, the relative GSH level in PEI-MONs treated cells (47.5%) was significantly lower than that in cells treated with PEI-MSNs (86.7%), suggesting that the tetrasulphide bond in PEI-MONs depleted intracellular GSH. mTORc1 pathway activation was then assessed by evaluating intracellular p-RPS6 levels [26] using flow cytometry in DC2.4 cells incubated with PEI-NPs for 24 h. As shown in Figure 5B, the relative MFI of Alexa Fluor 647 conjugated p-RPS6 antibody for cells incubated with PEI-MONs was significantly higher than that in the PEI-MSNs group (2.6 times). The same trend was observed using immunocytochemistry, with PEI-MON-treated cells (Figure 5D) exhibiting elevated p-RPS6 compared to the PEI-MSNs group (Figure 5E) and the PBS control (Figure 5C), consistent with enhanced mTORc1 activation. These results are consistent with tetrasulphide bonds of PEI-MONs promoting GSH depletion, leading to activation of the mTORc1 pathway and enhanced translation of pDNA-encoded proteins.

The pDNA transfection performance was further studied using pDNA encoding OVA in DC2.4 cells. OVA expression levels in cell lysates after OVA pDNA transfection were quantified by ELISA. Similar to EGFP pDNA transfection, OVA levels in pDNA/PEI-MON-treated cells were significantly higher than that of pDNA/PEI-MSNs and pDNA/lipofectamine groups (Figure 6A). This observation indicated enhanced OVA pDNA transfection by MONs with tetrasulphide bonds, and that our designed MONs can be used to deliver different pDNA into DCs. To further investigate GSH depletion as the likely mechanism for enhanced efficacy of MONs, the EGFP pDNA transfection assay was performed in the HEK293T cell line that has relatively low GSH levels [31]. Flow cytometry results showed no significant difference in positive cell percentage and MFI for pDNA transfection between PEI-MONs and PEI-MSNs in these cells, which were both lower than that observed with lipofectamine. The above results suggested that the enhanced pDNA transfection induced by GSH depletion was more pronounced in cells with high GSH levels (for example, DC2.4 cells) than in cells with low GSH levels (for example, HEK293T cells).

## 4. Conclusions

In conclusion, the impact of tetrasulphide bonds within MONs on pDNA transfection has been studied. Compared to inorganic MSNs, tetrasulphide-bridged MONs with GSH depletion capability can enhance activation of the mTORc1 pathway, consequently upregulating translation of pDNA-encoded mRNAs in high-GSH DCs. This enhancement is not apparent in low-GSH cells such as HEK293T cells, further supporting our proposed mechanism. Tetrasulphide bond-mediated pDNA transfection provides a foundation for the rational design of next-generation pDNA delivery systems to DCs.

## Figures and Tables

**Figure 1 pharmaceutics-15-01013-f001:**
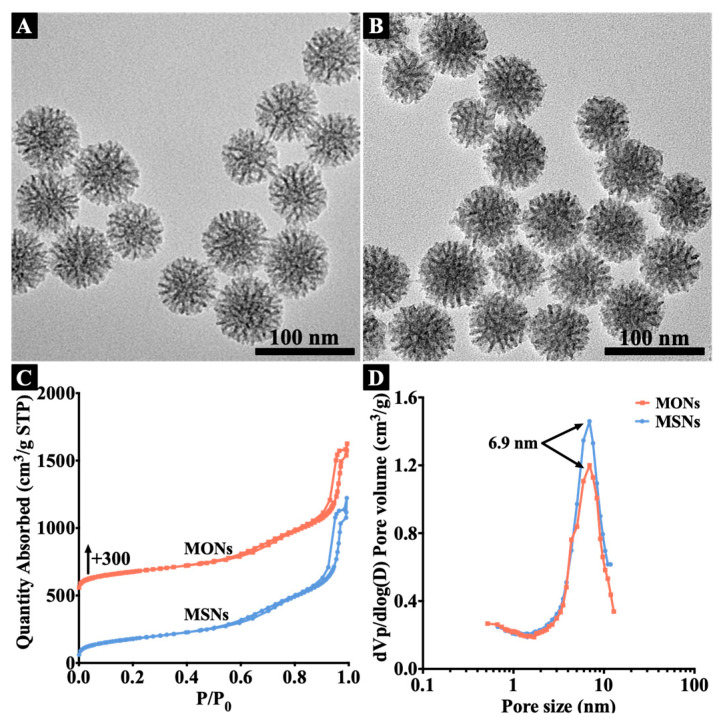
TEM images of MONs (**A**) and MSNs (**B**). Nitrogen sorption isotherms (**C**) and pore size distribution curves (**D**) of MONs and MSNs.

**Figure 2 pharmaceutics-15-01013-f002:**
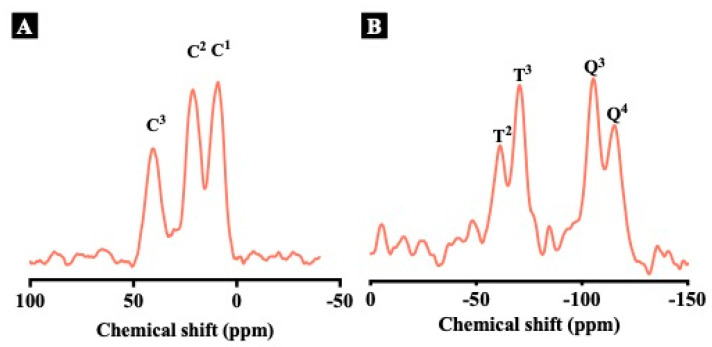
Solid–state ^13^C CPMAS NMR spectrum (**A**) and ^29^Si MAS NMR spectrum (**B**) of MONs.

**Figure 3 pharmaceutics-15-01013-f003:**
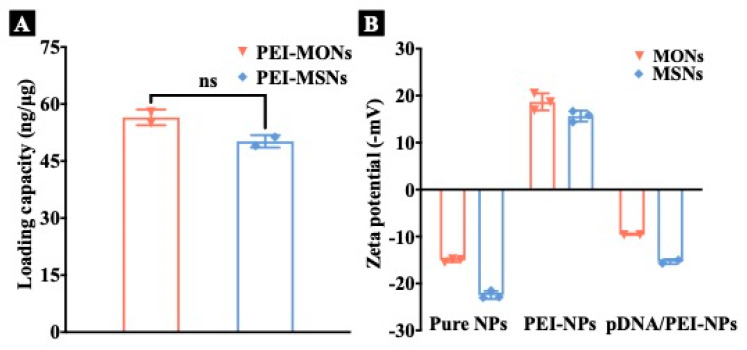
(**A**) EGFP pDNA loading capacity of PEI-NPs. (**B**) Zeta potential of pure NPs, PEI-NPs before and after pDNA loading. The results were analysed by unpaired *t*-test, ns: *p* ≥ 0.05.

**Figure 4 pharmaceutics-15-01013-f004:**
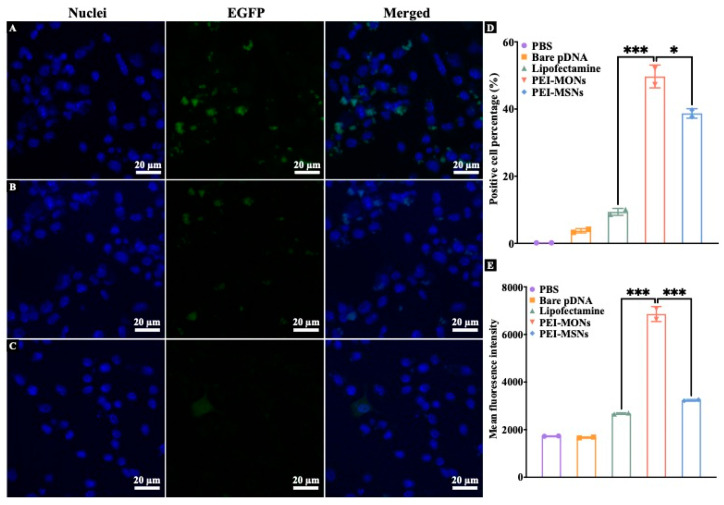
Confocal images of DC2.4 cells incubated with EGFP pDNA/PEI-MONs (**A**), EGFP pDNA/PEI-MSNs (**B**), and EGFP pDNA/lipofectamine (**C**). Flow cytometry analysis of EGFP positive cell percentage (**D**) and MFI (**E**) of DC2.4 cells incubated with pDNA/PEI-NPs complex for 48 h. The results were analysed by one way ANOVA multiple comparison, *: *p* < 0.05, ***: *p* < 0.001.

**Figure 5 pharmaceutics-15-01013-f005:**
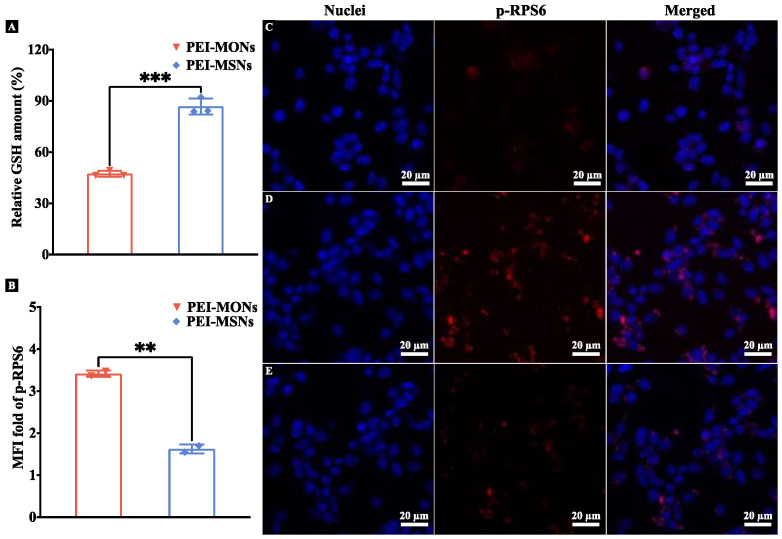
(**A**) Relative GSH levels in DC2.4 cells after incubation with PEI-NPs for 12 h. (**B**) Relative MFI of p-RPS6 in DC2.4 cells after incubation with PEI-NPs for 24 h. Confocal images showing p-RPS6 levels in DC2.4 cells incubated with PBS (**C**), PEI-MONs (**D**), and PEI-MSNs (**E**) for 24 h. The results were analysed by unpaired *t*-test, **: *p* < 0.01, ***: *p* < 0.001.

**Figure 6 pharmaceutics-15-01013-f006:**
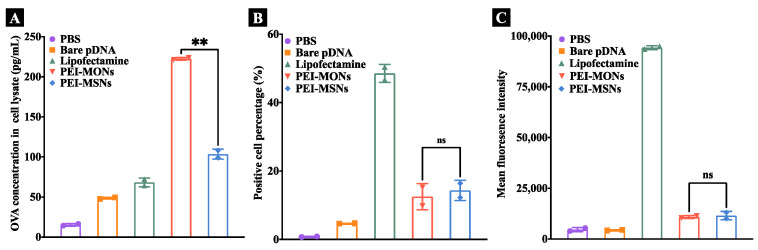
Relative OVA expression levels (**A**) as determined by an OVA-specific ELISA kit in lysates of DC2.4 cells treated with pDNA/PEI-NPs for 48 h. Flow cytometry analysis of EGFP positive cell percentage (**B**) and MFI (**C**) of HEK293T cells incubated with pDNA/PEI-NPs complex for 48 h. The results were analysed by one way ANOVA multiple comparison, ns: *p* > 0.05, **: *p* < 0.01.

## Data Availability

Not applicable.

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
