# Peer review of "Mesoporous Organosilica Nanoparticles with Tetrasulphide Bond to Enhance Plasmid DNA Delivery"

_pharmaceutics, 2023, doi:10.3390/pharmaceutics15031013_

Round 1
Reviewer 1 Report
Although the findings that tetrasulphide bridged MONs for pDNA transfection in Hela (cancer) cells and tetrasulphide bridged MONs enhance pDNA transfection efficacy have been reported, these studies lack relevant studies on the effect of MON structural difference on pDNA transfection. In the present manuscript, the effect of tetrasulphide bonds within MONs on pDNA transfection and translation efficacy is investigated. The findings in this manuscript may provide a new design principle of nano delivery systems where the pDNA delivery to DCs is important. To better facilitate the reader's understanding of this study, the following suggestions should be considered.
1. It is suggested to illustrate the main reactions involved in the preparation of nanoparticles, DNA adsorbed by nanoparticles.
2. It is suggested to using diagrams illustrating the differences between MSNs and MONs in GSH depletion capability and the related biological activities.
3. It is best to give direct experimental evidence that tetrasulphide bonds is altered after reaction by GSH.
Author Response
General comment: Although the findings that tetrasulphide bridged MONs for pDNA transfection in Hela (cancer) cells and tetrasulphide bridged MONs enhance pDNA transfection efficacy have been reported, these studies lack relevant studies on the effect of MON structural difference on pDNA transfection. In the present manuscript, the effect of tetrasulphide bonds within MONs on pDNA transfection and translation efficacy is investigated. The findings in this manuscript may provide a new design principle of nano delivery systems where the pDNA delivery to DCs is important. To better facilitate the reader's understanding of this study, the following suggestions should be considered.
Response: We thank Reviewer 1 very much for the positive comment. We have carefully considered your comments and revised the manuscript accordingly.
Comment 1: It is suggested to illustrate the main reactions involved in the preparation of nanoparticles, DNA adsorbed by nanoparticles.
Response: We thank Reviewer 1 for your constructive suggestion. The illustrations below for the reactions involved in nanoparticle preparation and DNA absorption on nanoparticles have been added in the revised manuscript (Page 5, 7).
“ In the reaction, both inorganic and organosilica precursors were hydrolyzed to provide Si-OH groups, which further assembled with the cationic head groups of the surfactant templates [25].” (Nanoparticle preparation, Page 5)
“Negative charged pDNA was absorbed with positively charged PEI via electrostatic attraction [30].” (DNA absorption on nanoparticles, Page 7)
Comment 2: It is suggested to use diagrams illustrating the differences between MSNs and MONs in GSH depletion capability and the related biological activities.
Response: We thank Reviewer 1 for the useful suggestion. The graphic abstract has been edited as attached to show the difference between MSNs and MONs in GSH depletion capability and translation regulation.

Reviewer 2 Report
This paper presents the interest of MONs containing tetrasulfide groups to induce pDNA transfection in cells thanks to a GSH depletion in DC cells.
This study complements the recently published paper ref 19.
The study seems properly conducted, but minor revisions should be applied before publication :
The MON with GSH-responsive tetrasulfide groups were first presented by J Shi (J. Am. Chem. Soc. 2014, 136, 16326−16334). This paper should be cited.
Furthermore, MONs or PMOs with disulfide or tetrasulfide groups have been studied, that degrade under GSH (See the works of L de Cola, JO Durand, and F Tamanoi). Is there any evidence of NPs degradation under GSH in the case of the present MONs ?
Minor comments :
- l 100 : The NPs are not dissolved, but dispersed
- l 225 : Round the pore diameter to 6.9 in both cases. Such a precision (lower than the size of an atom) makes no sense given the BJH curve with very few points.
- l 249 : Use 10 kDa instead of 10 K for the molecular weight of polymers.
- l 298 : isn't it pDNA/PEI-MSNs here, instead of MON ?
Author Response
General comment: This paper presents the interest of MONs containing tetrasulfide groups to induce pDNA transfection in cells thanks to a GSH depletion in DC cells. This study complements the recently published paper ref 19. The study seems properly conducted, but minor revisions should be applied before publication.
Response: We thank Reviewer 2 very much for the positive comment. We have carefully considered all your comments and revised them accordingly.
Comment 1: The MON with GSH-responsive tetrasulfide groups were first presented by J Shi (J. Am. Chem. Soc. 2014, 136, 16326−16334). This paper should be cited.
Response: We thank Reviewer 2 for the useful suggestion. This paper has been cited as ref 19 in the revised manuscript (Page 2).
“In particular, tetrasulphide bonds of MONs have been applied to deplete glutathione (GSH) [19] ”
Comment 2: Furthermore, MONs or PMOs with disulfide or tetrasulfide groups have been studied, that degrade under GSH (See the works of L de Cola, JO Durand, and F Tamanoi). Is there any evidence of NPs degradation under GSH in the case of the present MONs?
Response: We thank Reviewer 2’s constructive comment. We performed a degradation test to characterize the MONs degradation behaviour under GSH. The methods, results and discussion below have been added in the revised manuscript (Pages 3, 6-7)
“2.5 GSH oxidation assay
1.5 mg of nanoparticles (MONs or MSNs) were dispersed in 1 mL of PBS buffer con-taining 10% FBS and various concentrations of GSH (1 mM and 10 mM) and stirred at 37 °C for 48 h. Then aliquots were taken and washed with PBS before TEM imaging.” (Method, Page 3)
Figure S3. Degradation test of MONs (A-B) and MSNs (C-D) tested at 1 mM (A, C) and 10 mM (B, D) GSH solution for 48 h in the presence of serum. See attached
“Next, the GSH responsive degradability of MONs and MSNs was studied at 1 mM and 10 mM GSH solutions in the presence of serum, equivalent to the intracellular conditions of normal cells and dendritic cells, respectively. As shown in Figure S3A, after incubation of MONs with 1 mM GSH solution, partially collapsed structures were observed while the spherical structure was mostly maintained. However, when incubated with 10 mM GSH solution, the structure of MONs was seriously destructed and even partially disintegrated into small debris after 48 h (Figure S3B). In comparison, MSNs remained the initial structures at 48 h in both GSH concentrations (Figure S3C-D). The results above showed the GSH-responsive degradability of MONs.” (Results and discussion, Page 6-7)
Comment 3: - l 100 : The NPs are not dissolved, but dispersed.
Response: We thank Reviewer 2’s kind comment. The word has been changed in the revised manuscript (Page 3).
“30 mg of nanoparticles (NPs, either MONs or MSNs) were dispersed through sonication in 10 mL of pH=10 water”
Comment 4: - l 225 : Round the pore diameter to 6.9 in both cases. Such a precision (lower than the size of an atom) makes no sense given the BJH curve with very few points.
Response: We thank Reviewer 2’s helpful suggestion. The pore diameters for both nanoparticles have been revised accordingly (Page 5).
Figure 1. Pore size distribution curves (D) of MONs and MSNs. See attached
“Both nanoparticles showed peaks centred at 6.9 nm, indicating similar mesopore sizes for MONs and MSNs.”
Comment 5: - l 249 : Use 10 kDa instead of 10 K for the molecular weight of polymers.
Response: We thank Reviewer 2’s good suggestion. The unit has been changed in the revised manuscript (Page 7).
“Both MONs and MSNs were modified with the branched PEI with a molecular weight of 10 kDa.”
Comment 6: - l 298 : isn't it pDNA/PEI-MSNs here, instead of MON?
Response: We thank Reviewer 2’s friendly suggestion. The term has been edited in the revised manuscript (Page 8).
“Moreover, cells treated with pDNA/PEI-MONs also showed the highest mean fluorescence intensity (MFI), which was 2.1 and 2.6 times higher than that of cells treated with pDNA/PEI-MSNs and pDNA/lipofectamine, respectively.”

Reviewer 3 Report
This manuscript introduces the mesoporous organosilica nanoparticles modified with tetrasulphide bond to enhance the plasmid DNA delivery.
The manuscript is very well written and the experiments are well conducted and explained.
I recommend the publication of the article after performing the following:
1- The authors should explain the role of the branched PEI in increasing the loading and enhancing the delivery of the pDNA even with few sentences.
2- The authors should add the nature of the error bars in all the relevant figures (figures 3-6)
3- Why did the authors use PBS as a blank for the UV spectrophotometric determination of pDNA in the supernatant and not the supernatant obtained after the centrifugation of plain nanoparticles? How was the used method sensitive?
Author Response
General comment: This manuscript introduces the mesoporous organosilica nanoparticles modified with tetrasulphide bond to enhance the plasmid DNA delivery. The manuscript is very well written, and the experiments are well conducted and explained. I recommend the publication of the article after performing the following:
Response: We thank Reviewer 3 very much for the positive comment. We have carefully considered your comments and revised the manuscript accordingly.
Comment 1: The authors should explain the role of the branched PEI in increasing the loading and enhancing the delivery of the pDNA even with few sentences.
Response: We thank Reviewer 3’s constructive suggestion. The role of the branched PEI in enhancing pDNA delivery has been added in the revised manuscript (Page 7).
“To enhance pDNA delivery, PEI with a highly positive charge and effective proton sponge effect was modified on nanoparticles for enhancing their pDNA loading and endosomal escape capacity [29]. Both MONs and MSNs were modified with the branched PEI with a molecular weight of 10 kDa.”
Comment 2: The authors should add the nature of the error bars in all the relevant figures (figures 3-6).
Response: We thank Reviewer 3’s kind suggestion. The nature of the error bars has been added to Figure 3-6 and Figure S6 in the revised manuscript (Page 7-10).
Comment 3: Why did the authors use PBS as a blank for the UV spectrophotometric determination of pDNA in the supernatant and not the supernatant obtained after the centrifugation of plain nanoparticles? How was the used method sensitive?
Response: We thank Reviewer 3’s helpful comment. The supernatant obtained after centrifugation of pure nanoparticles was used to replace PBS as a blank for determining pDNA loading capacity. The results obtained were similar to that using PBS as the blank, which were 56.3 and 50.6 ng μg−1 for PEI-MONs and PEI-MSNs, respectively, thus PBS was selected as the blank.

Reviewer 4 Report
The authors investigated the ability of tetra sulfide bridged mesoporous organosilica nanoparticles (MONs) for pDNA transfection performance in DC cell lines.
The overall study is interesting, and a good read. The paper can be considered for publication.
Author Response
General comment: The authors investigated the ability of tetra sulfide bridged mesoporous organosilica nanoparticles (MONs) for pDNA transfection performance in DC cell lines. The overall study is interesting, and a good read. The paper can be considered for publication.
Response: We thank Reviewer 4 very much for the positive and kind comment.